# A Versatile Suspended Lipid Membrane System for Probing Membrane Remodeling and Disruption

**DOI:** 10.3390/membranes12121190

**Published:** 2022-11-25

**Authors:** Achinta Sannigrahi, Vishwesh Haricharan Rai, Muhsin Vannan Chalil, Debayani Chakraborty, Subrat Kumar Meher, Rahul Roy

**Affiliations:** Department of Chemical Engineering, Indian Institute of Science, Bangalore 560012, Karnataka, India

**Keywords:** suspended bilayer, pore-forming toxin, Cytolysin A, virus fusion, pore formation

## Abstract

Artificial membrane systems can serve as models to investigate molecular mechanisms of different cellular processes, including transport, pore formation, and viral fusion. However, the current, such as SUVs, GUVs, and the supported lipid bilayers suffer from issues, namely high curvature, heterogeneity, and surface artefacts, respectively. Freestanding membranes provide a facile solution to these issues, but current systems developed by various groups use silicon or aluminum oxide wafers for fabrication that involves access to a dedicated nanolithography facility and high cost while conferring poor membrane stability. Here, we report the development, characterization and applications of an easy-to-fabricate suspended lipid bilayer (SULB) membrane platform leveraging commercial track-etched porous filters (PCTE) with defined microwell size. Our SULB system offers a platform to study the lipid composition-dependent structural and functional properties of membranes with exceptional stability. With dye entrapped in PCTE microwells by SULB, we show that sphingomyelin significantly augments the activity of pore-forming toxin, Cytolysin A (ClyA) and the pore formation induces lipid exchange between the bilayer leaflets. Further, we demonstrate high efficiency and rapid kinetics of membrane fusion by dengue virus in our SULB platform. Our suspended bilayer membrane mimetic offers a novel platform to investigate a large class of biomembrane interactions and processes.

## 1. Introduction

Lipid membranes not only protect cellular integrity and define structure, they have a key role to play in most physiological processes including cell-to-cell communication, intercellular and intracellular control of micro-environments, and metabolism [1,2,3]. Although membranes have been intensively studied over the last 60 years [4], several long-standing questions regarding their structural organization and the interaction with and between embedded components at the molecular level remain unanswered. Biological membranes display a very complex composition of lipids and proteins. Further, they vary tremendously in local composition and architecture even within the same eukaryotic cell [3]. This intrinsic complexity of biological membranes renders their systematic investigation challenging. Therefore, researchers have generated synthetic models capable of mimicking the properties and functions of the cellular membranes. These simple models not only retain the essential lipid bilayer structure by self-assembly, but also offer freedom to vary the composition and membrane properties, such as interfacial tension, phase segregation, and packing. This allows one to investigate and infer mechanistic details of biomembrane phenomena decoupled from gene expression and cytoskeletal interference [5].

One of the most popular model membrane systems is the unilamellar vesicle (single bilayer lipid membrane enclosing aqueous compartments) mimicking the bilayer configuration all major cellular membranes. Due to their high curvature, small unilamellar vesicle (SUV) and large unilamellar vesicle (LUV) are commonly employed for vesicle studies but not considered appropriate mimics for the plasma membrane. Giant unilamellar vesicles (GUV) display large variation in their size distribution that is difficult to control. Further, imaging of molecular processes on GUVs is harder due to fluctuations of the membrane bilayer. This has led to the development of many planar membrane systems including the black lipid membrane (BLM), supported lipid bilayer membrane (SLBM), air-stable lipid bilayer membrane (as-LBM), hybrid lipid bilayer membrane (h-LBM), and polymer-cushioned lipid bilayer membrane (pc-LBM) [3,6]. Each approach solves some challenges at the cost of others pertaining to the ease of formation, yield, membrane stability and lifetime. Planar lipid membranes have been widely used for electrical detection of the activity of reconstituted ion channels [7] but these membranes are mechanically poor in stability as they are usually spread over large orifices and are not suitable for monitoring transmembrane transport for a long time. Supported lipid bilayer (SLB) membranes are commonly employed to probe interactions using microscopy but access to SLB is limited from the top of the bilayer. SLBs are prepared by diverse techniques including vesicle fusion and solvent-assisted lipid bilayer method [8], Langmuir Blodgett trough [9] and painting method [10]. SLB membranes are widely used due to their ease of formation, stability and ease of microscopic investigation [8,11,12,13]. However, these advantages come at the expense of reduced mobility of the lipids and embedded components imparted by interactions with the underlying surface [11]. The entrapped thin layer of water underneath the membrane [8] is also poorly understood and interferes with transmembrane protein function. To overcome these challenges, modifications to SLBs have been introduced, where the lower leaflet is decoupled from the solid substrate with a polymer cushion of Polyethylene Glycol (PEG) or Dextran [14] or by tethering the membrane to the substrate with linkers with polymer cushion [15]. With increasing polymer cushion length, there is an observed increase in the mobility as well as the hydration of the membrane interface. However, with an increase in hydration, the polymer cushion swell, which creates defects in the membrane [16]. The PEG polymer also leads to crowding in the membrane beyond a certain concentration, which restricts the movement of the lipids on the membrane and generates undesired defects [11,17]. In many applications, flat membranes that cannot undertake large deformations can be disadvantageous and may not reproduce the biophysical processes faithfully. For example, researchers employ SLBs for membrane fusion that might face a large barrier due to extensive surface interactions between the SLB and underlying substrate. Similarly, membrane fusion in case of many enveloped viruses occurs in significantly curved endosome membranes. The curvature stresses play a role in the thermodynamics of membrane fusion [18]. Dedicated protein machineries that control the fusion of viral membrane [19] might be designed to operate under defined membrane curvatures and would display poor and heterogeneous activity. Therefore, there is a need to develop defined curved free standing membranes topologies for study of many membrane processes.

To alleviate this challenge, suspended membranes spanning nanofabricated channels have been developed [20]. They are particularly attractive for investigating the detailed function of isolated membrane proteins in a high-throughput fashion [20]. These systems generally exhibit a much higher sealing resistance than a cell membrane patch, which is advantageous for single-channel recordings [21,22]. Free-standing bilayer membranes, including BLMs, display unrestricted lipid mobility [23] and enable association of the membrane biomolecules within or on both sides of the membrane [22]. Free standing membranes are usually generated using silicon [20,24,25,26,27,28,29,30,31] or aluminum oxide wafers [32]. There are several disadvantages to using silicon-based devices, including the low resistivity of silicon, the high dielectric loss, and the high shunt capacitances that result from using thin insulating dielectrics [33]. This leads to increased electrical noise and restricted bandwidth. In the case of Si/SiO_2_ nanodevices, the membranes formed over the porous substrate can come into contact with the rim of the microwells and due to the heavy impact with the rim of the microwells the pore spanning suspended membranes may rupture [34]. Furthermore, the fabrication of silicon-based devices is time-consuming and expensive, the resulting devices are fragile, and the integration of microfluidics can be complicated. On the other hand, polymeric nanostructures have the potential to overcome many aforementioned disadvantages. They are electrically insulating, robust, and can be fabricated using rapid, low-cost methods that are amenable to the simple large-scale production of microfluidic systems [33].

Here, we present suspended lipid bilayer (SULB) membranes on etched microwell cavities in polycarbonate track etched (PCTE) substrate as a platform for the analysis of membrane pore formation and membrane fusion events. The PCTE substrate is coated with Polyvinylpyrrolidone (PVP), a water-soluble polymer made from its monomer N-vinylpyrrolidone. These polymers are highly hydrophilic [35]. The polymer substrate also prevents interactions with the rim and cushions the membrane. Moreover, due to availability of PCTE PVP coated hydrophilic substrates with different microwell diameters, we can identify the effect of suspended membrane area on its stability and configuration.

Further, we demonstrate that our PCTE-based SULB platform can be used for membrane pore formation analysis by pore-forming toxins (PFTs). Using different lipid compositions, we show that the ternary (containing Cholesterol and Sphingomyelin) phase separated membrane facilitates the pore-forming activity of ClyA, an alpha pore forming toxin (PFT). Further, we examine the effect of membrane curvature and lipid phase segregation on PFT action and trans-bilayer lipid flipping induced by the PFT. As a complementary proof-of-concept demonstration, we monitor membrane fusion of the dengue virus (DENV) on suspended lipid bilayers. Overall, we demonstrate that our novel robust, easy to use, and versatile SULB system can be used as a lipid membrane mimic for numerous biophysical studies.

## 2. Results and Discussion

### 2.1. Suspended Bilayer Membranes on PCTE Substrates

We prepared the SULB by rupturing the GUVs on the PCTE substrate surface. The GUVs were prepared using gel-assisted GUV preparation method [36] and then characterized with phase contrast microscopy. For the imaging of prepared GUVs, we used sucrose and glucose inside and outside of the vesicle, respectively. The refractive index difference between the two sugars was generates high phase contrast in the GUV images (Appendix A). Using this method, we recovered a relatively high yield of the uniformly sized vesicles (10.53 ± 6.03 μm). As the size of these GUVs was greater than the pore size of the porous substrate, they could be used directly for the preparation of lipid bilayer membrane over the porous substrate by vesicle rupture and membrane spreading.

PCTE substrate was placed on a cleaned cover glass before addition of GUVs and their rupture during incubation (Figure 1). AFM images help in confirming the suspended lipid bilayer membrane formation over the porous PCTE substrate. The average depth measured for the membrane-free microwell 126 ±18 nm drops upon membrane coverage to 20–70 nm (Figure 2A,B). The spread of ruptured GUVs over the substrate, thus forming the lipid bilayer membrane over the track-etched microwells, was discerned by the reduction of ~50–100 nm in axial depth and a reduced contrast in the AFM image (Figure 2A,B). The deflection of suspended membrane due to bending observed with AFM is consistent with larger spring constant of microwell spanning membranes compared to the membrane region over the PCTE substrate [37,38]. We hereby define curvature, as the ratio of maximum depression at the center of an individual membrane suspension over a pore to the radius of the pore. When we compare the curvature of suspended membranes with various lipid compositions, ternary mix of POPC:SM:CHOL (1:1:1) displays the smallest curvature (0.047 ± 0.014) followed by binary POPC:CHOL (1:1) (0.068 ± 0.017) and POPC (0.10 ± 0.01) membranes (Figure 2B–D). POPC remains in liquid disordered phase at room temperature but the addition of cholesterol increases the order (Lo domain formation) in binary membranes and addition of sphingomyelin further enhanced the lipid order [39]. This increased domain order results increased membrane stiffness, contributing to reduced bending. We also note increased membrane surface roughness in ternary SULB that further suggests phase segregation to Lo/Ld phases (Figure 2B(iv)). The narrow margin of the curvatures is indicative of the uniformity in membrane bending throughout the pore spanning membrane (Figure 2D). Therefore, our SULB platform recapitulates bilayer lipid composition mediated curvature and stiffness effects in the suspended membranes.

Next, to visualize the SULB by fluorescence microscopy, we prepared DiI (DiI C18(5);1,1′-dioctadecyl-3,3,3′,3′tetra-methyl-indo-di-carbocyanine) labeled SULB platform (Appendix A). The photobleaching of DiI is slower in the supported compared to the suspended membranes possibly due to limited excitation of the suspended segment of the bilayer (Appendix A). This could arise from the direct interaction with the underlying substrate and the presence of pinning sites leading to restricted mobility in the supported segment of the bilayer. The intensity variations corresponding to different microwell sizes show an expected trend wherein the intensity increases with the increase in the microwell size since the microwell size limits the bending and number of DiI molecules (Figure 2E). Due to the illumination of the bilayer from the bottom of the substrate, the DiI in the lipid bilayer not on the microwells is not completely excited providing enhanced contrast.

To examine diffusion of lipids in the suspended membranes, the mobility of Liss-Rhodamine-DOPE lipid embedded in the binary and ternary bilayer was evaluated by FRAP. Figure 3A shows FRAP measurement in a ternary lipid bilayer suspended over a microwell with diameter of 1 µm. The bleached area was set to match the suspended part of the membrane (1 µm diameter). Fluorescence recovery rates were considerably slow and complete recovery of pre-bleached fluorescence is not achieved (Figure 3B,C). From the fluorescence recovery curves, the apparent diffusion coefficients were estimated to be D = 0.0049 µm^2^/s for binary composition and D = 0.0037 µm^2^/s for ternary system (Figure 3D). The slow recovery rates in the suspended region further underlines the local confinement of the suspended membrane across the PCTE rim supported part of the bilayer due to the significantly slower lipid exchange between the supported and the suspended regions. Previous reports indicate a similar restricted diffusion behavior of Rhodamine DOPE in the suspended bilayer segments [40]. It has been shown that Lo domain preferably stays in the supported part and the pore rims [41]. In contrast, the Ld phase is present in the suspended part. We suspect that the high concentration of cholesterol can lead to local domain formation in our SULB and these domains give rise to local confinement, which could further lead to lower diffusivity of Rh DOPE in the suspended section. Moreover, the lower diffusivity in the ternary bilayer in comparison to binary lipid membrane composition is consistent with the increased internal rigidity in the phase separated ternary bilayer.

### 2.2. Stability of PCTE-SULB Membranes

To examine the membrane stability, Sulforhodamine B (SRB) dye was entrapped in the microwells capped by a bilayer allowing the fluorescence imaging of individual cavities. A significant intensity difference between the supported and suspended part of our SULB was observed due to the entrapped dye (Figure 4A and Appendix A). To measure the stability of the membrane, we imaged the cavities at different intervals of time over 66 h and quantified the fraction of intact microwells based on SRB fluorescence. It is to be noted that for each occasion, we performed multiple gentle washing of the membrane to remove the leaked dye from the ruptured micro wells before imaging. Our SULB membranes on 1 μm microwells was significantly stable as ~100% of the sealed population as well as fluorescence intensity per well was retained up to 66 h (Figure 4A). The narrow range of the SRB intensity distribution also depicted the significant stability of our SULB system. In contrast, we observed significantly reduced stability of 3 μm SULB system as observed as loss of dye by 18 h and final leakage of 50% cavities by 66 h (Figure 4B). The correlation between the stability of the membrane and pore size has been established by various groups and they have shown that higher apertures corresponds to lower stability of the membrane [42,43]. It is also well-known that the substantial reduction in pre-stress of the bilayer membrane results in lowers the membrane rupture frequency [37]. We speculate that due to the bending of the lipid bilayer on the cavity of higher aperture (3 μm), the hydrophilic interactions between the rim surface and bilayer head region weaker than the hydrophobic interaction in the bilayer and this difference in turn can result in the instability of the membranes on larger cavities.

### 2.3. Cytolysin A Pore Formation in Ternary SULB Membranes

Although, it is known that cholesterol helps the assembly pathway of ClyA (a bacterial pore forming toxin) by stabilizing the intermediate structure, the role of phase separated membrane on ClyA pore forming activity is not explored yet [44]. Since recent reports revealed that the membrane permeabilizing activity of Equinatoxin II, an α-PFT, depends on the presence of sphingomyelin and lipid phase coexistence [45], it is essential to study the role of phase separation in the membrane on the toxin activity. Another important parameter that can regulate the pore-forming activity of ClyA is the membrane curvature that plays a crucial role in protein-lipid binding and pore formation. The effect of the curvature of the membrane has not yet been probed for its impact on the pore-forming activity of ClyA. Here, we used the model SULB system to investigate the pore-forming activity of the toxin further. Membranes comprising binary and ternary lipid mixtures were utilized as the model suspended bilayers for understanding the effect of phase separated membrane on the pore-forming activity of ClyA. As the diameter of the PCTE-substrate ‘well’ strongly influences the curvature of the suspended membrane, we used substrates of different well-sizes (0.6 and 1 µm) to explore the correlation between membrane pore size and the pore-formation activity. The temporal release of the trapped SRB dye from the micro cavities after the addition of protein reports on the PFT-induced membrane pore formation (Figure 5A). This dye release was quantified by the temporal reduction in the fluorescence intensity of each well along with the frequency distribution of the SRB dye filled intact microwells.

We observed significant and rapid drop in SRB signal upon ClyA addition (Figure 5B). Our results showed that the ClyA pore formation in SULB was faster for ternary membrane in both PCTE microwells compared to the binary SULB (Figure 5B–D, Appendix A). Next, we evaluated the half-life of pore formation for each system. Interestingly, we found that for pore sized PCTE substrates of 1 μm and 600 nm, a significant drop in half-life happened as we moved from binary to ternary lipid compositions used for the making of SULB platform (Appendix A). Thus, our data suggested that lipid composition appreciably affects the pore forming toxin, ClyA activity. Since the dodecameric pore of ClyA can traverse single bilayer as evident from our membrane docked structure obtained from OPM (Orientations of proteins in membranes) server [46] and rapid dye leakage is detected after addition of ClyA, we argue the unilamellarity of our SULB system (Figure 5E). We compared the ClyA induced pore formation in SULB with published ensemble assays [44]. The average dye leakage half-life obtained for bulk assay was found to be smaller than our SULB system. The fluorescence measurements of individual cavities in only one experiment yields high statistical information from more than hundred cavities simultaneously and thus our system became an exciting platform for membrane disruption assays.

### 2.4. Trans-Bilayer Lipid Movement in SULB

The trans-bilayer movement of glycerophospholipids, popularly referred to as flip-flop in membranes is paramount to cellular function (shown schematically in Figure 6A,B). It is well known that the number and distribution of the lipid species present on the extracellular and luminal leaflets are regulated to control cell signaling events, localization of membrane components and to affect membrane shape changes [47]. Therefore, lipid flip-flop plays an important role in cell homeostasis and intercellular signaling.

Here, we monitored the spontaneous lipid flip-flop motion in a binary lipid bilayer with the SULB platform after the addition of ClyA. We adapted the sodium dithionite (Na_2_S_2_O_4_) reduction strategy [48,49] with NBD-PC labeled membranes to probe the trans-bilayer movement (Figure 6B). The temporal dynamics were measured by the fluorescence quenching of the NBD probe upon sodium dithionite addition. We measured the time dependent loss of NBD intensity for both supported part and suspended part of our SULB platform in absence and presence of ClyA (Figure 6C–E). The control experiment was carried out on NBD SULB without sodium dithionite to check for the effect of photobleaching. An insignificant change in NBD Fluorescence signal with time that eludes the effect of photobleaching was observed (Appendix A). We found considerable difference in reduction kinetics of NBD fluorescence in supported and suspended part (Figure 6C, inset). This result further suggests the restricted lateral mobility in the supported part also restricts the trans-bilayer movement. NBD reduction rate on the SULB is significantly enhanced upon ClyA addition (Figure 6E). We observed that the half-life of the reduction for the suspended part in absence of ClyA was estimated to be 0.2 min by fitting the decay curve to a single-exponential function (Figure 6E, inset). The value was found to be higher than the reported value (ca. 0.1 min) at an identical dithionite concentration [48]. This discrepancy can be attributed to the presence of high cholesterol concentration, which presumably decreases the effective concentration of the S_2_O_4_^2−^ ions on the membrane surface due to the increased stiffness of the bilayer. Interestingly, a significantly elevated reduction rate after addition of ClyA was observed indicating an enhancement in trans-bilayer dynamics in the SULB (Figure 6D,E, inset). It is to be noted that after the addition of ClyA, the boundary between kinetic traces of supported and suspended part was diminished (Figure 6E) further inferring a plausible ClyA driven lipid mixing in SULB system.

### 2.5. Virus Membrane Fusion in PCTE-SULB

The dengue virus enters the cell by receptor-mediated endocytosis followed by fusion within the cell compartment triggered by mild acidic pH within the lumen of the endosome [50]. In the convex surface of the cells, the virus binds to the clathrin-coated pits where the virions roll over, and subsequently, the clathrin-coated pits invaginate the plasma membrane, which is transported away from the plasma membrane and the pit are released [51]. For this, our suspended lipid bilayer membrane system would be an ideal platform to study the release of the content inside. The concave curvature of the suspended bilayer is similar to an endosome, providing an excellent model for studying viral membrane fusion kinetics. To test the viral membrane fusion, we first generated and purified DENV. We verified them by immunocytochemistry with anti-envelope antibody and the negative TEM imaging (Appendix A). The DENV membrane fusion was followed by introducing a lipid binding dye, DiD into the viral envelope [52]. Earlier reports suggested that DiD spontaneously inserts into the lipid bilayer in self-quenching concentrations, upon membrane fusion, the diffusion and dilution of DiD in the target membrane would result in dequenching of the signal followed by lateral dissipation of the fluorescence [52]. To monitor the DENV membrane fusion process with physiologically relevant membrane [53], a lipid mixture of POPC:SM:CHOL:POPG of equimolar concentration was used to form GUVs for spreading on SULB on 1 μm microwells. To identify the microwells, SRB dye was entrapped in the suspended bilayer allowing the suspended bilayer and the virus to co-localized in imaging (Figure 7A). As anticipated, there is poor fusion of DiD labeled DENV on SULB at physiological pH (7.5) as virus puncta are largely diffraction limited (Figure 7B). On the other hand, after the addition of acetate buffer (pH 5.5), the DiD diffuses throughout the microwells of the SULB along with an associated efflux of SRB dye from the SULB cavities (Figure 7C).

This is expected because acidification triggers membrane fusion mediated by the conformational rearrangement of the envelope protein domains from a pre-fusion dimer to a post-fusion trimer [54,55]. The broad Gaussian distribution along with the dequenched DiD signal signifies the fusion of DENV envelop upon SULB membranes (Figure 7C). Importantly, the fusion of individual viruses can be followed upon acidification as characteristic dequenching of DiD (Figure 7D,E). This demonstrates that suspended lipid bilayers composed of physiologically relevant lipids can be used to study the kinetics of membrane fusion.

## 3. Conclusions

Here, we present the track etched-polymeric filter membranes to establish a free-standing lipid bilayer system. We chose the PVP coated PCTE membrane for fabrication of the microwell spanning bilayer membranes in order to provide required stability and accessibility. The fabrication of our SULB membrane is simple and overcomes the complexity of using photolithography on silicon substrates. Our SULB system is an exceptionally stable system although the stability is dependent on the substrate microwell dimensions. The SULB system is versatile and amenable to sensitive fluorescence microscopy and related assays. We show this by measuring lipid diffusion (FRAP), membrane curvature and heterogeneity (AFM), membrane rupture (dye leakage), lipid flip-flop (NBD-lipid quenching), and membrane fusion (DiD dequenching). We show that Cytolysin A induced pore formation is enhanced on phase separated membrane. Moreover, ClyA is responsible for increased lipidic movement across the leaflets of the bilayer. Since our SULB system is a better mimic for the endosome membranes, we demonstrate that it recapitulates the low pH induced dengue virus fusion and displays faster kinetics than observed with supported bilayers. Thus, our suspended bilayer membrane mimetic provides an exciting new platform to study membrane protein interactions.

## 4. Materials and Method

### 4.1. Materials

The PCTE membrane filters (13 mm diameter) were purchased from STERLITECH and was used without further modification as a support for the SuLB. Sphingomyelin (Brain, Porcine) (SM), 1-palmitoyl-2-oleoyl-glycero-3-phosphocholine (POPC) and 1-palmitoyl-2-oleoyl-sn-glycero-3-phospho-(1′-rac-glycerol) (POPG) were purchased from Avanti Polar Lipids Inc. (Alabaster, AL, USA). The 1,1′-dioctadecyl-3,3,3′,3′-tetramethylindocarbocyanine perchlorate (“DiI”; DiIC_18_(3)), DiIC18(5) solid (1,1′-Dioctadecyl-3,3,3′,3′-Tetramethylindodicarbocyanine, 4-Chlorobenzenesulfonate Salt) (DiD) and NBD-PC (1-palmitoyl-2-{12-[(7-nitro-2-1,3-benzoxadiazol-4-yl)amino]dodecanoyl}-sn-glycero-3-phosphocholine) were purchased from Invitrogen (Eugene, OR, USA). Sulforhodamine B acid chloride and 18:1 Liss Rhod PE dye was purchased from Merck. These were used without further purification. All other necessary chemicals, including salts were obtained from Aldrich (St. Louis, MO, USA) and Merck (Mumbai, India).

### 4.2. GUV Formation

The GUV preparation was carried out by a gel assisted vesicle formation method [56]. A total of 5% *w*/*w* of Poly Vinyl Alcohol (PVA) solution was prepared. A total of 800 μL of 5% *w*/*w* PVA were spread over a petri-dish and was kept in over at 37 °C for 30 min. It eventually forms a thin polymer film of PVA on the petri-dish. 1-palmitoyl-2-oleoyl-sn-glycero-3-phosphocholine (POPC) (Avanti Lipids, Birmingham, AL, USA) and Cholesterol were dissolved in chloroform and the solution of 1 mg/mL of lipid mix containing required lipid compositions were made separately. The organic solvent was evaporated by vacuum desiccation for one hour, leaving behind a thin lipid film. Then, the lipid film was hydrated by the addition of 1 mL sucrose (0.1 M) and incubated for 30 min at room temperature. The petridish was stirred gently in order to dislodge the hydrated vesicles. Generally, the hydration of the GUVs is accomplished within 20 min. Once the hydration was done, the GUV suspension was transferred to a micro centrifuged tube using a pipette. Then, the sucrose containing GUVs were diluted with a glucose solution in the ratio of 1:3. This creates a difference in the refractive index which helps in the visualization of GUVs with phase contrast microscopy [56,57]. Coverslips (24 × 50 × 0.13 mm^3^, VWR International, Radnor, PA, USA) were cleaned thoroughly and an O ring is adhered to the coverslip in order to make a compartment for the GUVs. The GUVs were visualized using a 100× objective (Numerical Aperture = 1.4, Ph3 annulus) on a Phase contrast microscope (Nikon, Tokyo, Japan). It is to be noted that for phase contrast microscopic imaging we used glucose and sucrose, while for other experiments we made the GUVs in 1X PBS.

PCTE substrate and Suspended bilayer formation: The PCTE substrates were purchased from STERLITECH (Auburn, WA, USA). For the generation of the PCTE substrates, a nonporous polycarbonate film is exposed to heavy ions in a particle accelerator, which leaves “tracks” as the polymer chains are ruptured and ionized, while the ions traverse across the depth of the substrate. The polycarbonate film is then “etched” in a bath containing warm NaOH to form the pores. Control of temperature, pH and etching time determines the pore dimensions in the PCTE.

Pore spanning suspended bilayers were formed for visualization with the fluorescence microscopy and the sulforhodamine B leakage experiment. The GUV solution was doped with 25nM DiD (DiIC18(5);1,1′-dioctadecyl-3,3,3′,3′tetra-methyl-indo-di-carbocyanine, 4-chlorobenzene sulfonate salt) for visualization in the fluorescence microscope. For the leakage experiments, 40 μL of 100 nM sulforhodamine B dye was trapped in the microwell of the pore spanning substrate and upon addition of GUV solution and incubation. For the preparation of suspended lipid bilayer, the Polyvinylpyrrolidone (PVP) coated continuous pore hydrophilic substrates were kept over a thoroughly cleaned coverslip (24 × 50 × 0.13 mm^3^, VWR International). A total of 1 mL sucrose hydrated GUV solution was pipetted over the surface of the continuous pore spanning substrate. Then, the GUV solution coated substrate was incubated for 30 min at 37 °C to trigger the vesicle rupture over the PVP coated hydrophilic surface of the substrate. Excess lipid materials were removed by carefully rinsing with de-ionized water.

### 4.3. Atomic Force Microscopy

The AFM imaging of the pore spanning suspended lipid bilayers were carried out in Park NX-10 AFM system (Santa Clara, CA, USA). A contact cantilever with a spring constant of 0.2 N/m is used. The AFM image of the blank porous substrate was first generated and was then compared with the samples, where GUV solution was incubated, to check the formation of pore spanning suspended lipid bilayer.

Expression and Purification of ClyA PFT: ClyA expressing plasmids were used to transform *E. coli* BL21 (DE3) competent cells as previously published [44]. Single colonies obtained were grown in Terrific Broth (Pronadisa) media at 37 °C, 180 rpm in a shaker incubator. As the OD600 of the culture reached approximately 2, protein expression was induced by addition of 500 μM of isopropyl thio-galactopyranoside (IPTG) for 12 h at 16 °C. Cells were lysed by sonication in a buffer containing 100 mM Tris–HCl (pH 8.0), 20 mM β-mercaptoethanol, 100 mM NaCl, 1 mM benzamidine, 2 mM PMSF (Phenylmethylsulfonyl fluoride), and 10% glycerol. It was then centrifuged at 30,000× *g* and the cell-free extract was interacted with nickel–nitrilotriacetic acid beads [58]. Beads were washed with buffer containing 100 mM Tris–HCl (pH 8.0), 20 mM β-mercaptoethanol, 500 mM NaCl, and 20 mM imidazole to remove non-specific proteins on the beads and was eluted in the same buffer but containing 300 mM imidazole. The purified protein was concentrated with an Amicon 8050 concentrator (Danvers, MA, USA) ultrafiltration unit that is equipped with a regenerated nitrocellulose filter (Millipore Corp., Bedford, MA, USA) with a molecular mass cut-off of 10 kDa. Gel filtration was carried out in buffer containing 100 mM Tris-HCl (pH 8.0), 20 mM β-mercaptoethanol, 100 mM NaCl and 10% glycerol on an AKTA fast protein liquid chromatography system (GE Healthcare, Chicago, IL, USA) using the Sepharose 12 column (GE Healthcare). The run was performed at 4 °C at a flow rate of 0.25 mL/min and three peaks were observed with the predominant peak corresponding to our monomeric fraction of the protein. The eluted fractions were verified on a 12% poly-acrylamide gel in the Tris-Glycine SDS buffer. Protein concentration was estimated by Bradford protein assay.

## 5. ClyA Pore Formation/Leakage Study on Suspended Lipid Bilayer

A total of 100 nM of Cytolysin A protein were prepared by diluting the stock in 1X PBS buffer (pH 7.4, phosphate buffer concentration of 0.01 M). Initially a sample having pore spanning suspended membrane and SRB dye trapped in the microwell, without Cytolysin A spread over it (blank sample) was fluorescently imaged and kept as a reference, in order to see the extent leakage caused by the PFTs. A total of 40 μL of Cytolysin A solution (100 nM) was then spread over the pore spanning suspended lipid bilayer to reach a final concentration of 40 nM and incubated at room temperature. The leakage of SRB was monitored with time to measure pore formation by Cytolysin A. For the measurement of ClyA induced membrane pore formation, we used binary (POPC:CHOL, 1:1) and ternary lipid systems POPC:CHOL:SM, 1:1:1).

### 5.1. Fluorescence Microscope Setup and Analysis

SRB dye (100 nM) entrapped PCTE SULB platform were measured on an inverted microscope (Olympus IX81, Tokyo, Japan). A laser of 532 nm (Sapphire; Coherent) was used to excite the SRB molecules on a custom-built objective-type total internal reflection microscope. A combination of 25.4- and 300-mm biconvex lenses (Thor Laboratories, Newton, NJ, USA) were used to expand the laser beam before a lens of focal length 150 mm was used to focus the (16-mW) laser beam on its back focal plane (BFP) of the objective (UAPON 100× OTIRF; Olympus, Tokyo, Japan). The laser spot at the BFP was translated away from the optical axis to achieve inclined illumination to reduce the background. Fluorescence emission from 80 μm × 40 μm area was collected by the objective and passed through a dichroic mirror (FF545/650-Di01-25 × 36; Semrock, West Henrietta, NY, USA) and a long-pass filter (BLP02-561R-23.3-D; Semrock, West Henrietta, NY, USA) before detection on an electron multiplying charge-coupled device (Andor ixon Ultra 897, Belfast, UK). Shutter (LS6; Vincent Associates, Rochester, NY, USA) was used to control the laser illumination time. Subsequently, all the images were registered with the negative control for stage-drift correction using SIFT algorithm [59]. Circular Hough transform [60] was applied to the registered and background-subtracted images for locating the wells on the negative control. The resultant intensity histograms were plotted with time as a heatmap, which demonstrated the pore forming activity by ClyA at different conditions.

### 5.2. Fluorescence Recovery after Photobleaching (FRAP) of Lipid Bilayers

We prepared our SULB platform using binary and ternary lipid compositions doped with lipophilic tracer Liss Rhodamine DOPE. The sample was imaged on a Leica STED SP5 confocal microscope with an Argon laser and the emission was separated and collected between 550 and 580 nm by adjusting the acousto-optical beam splitter. Fluidity of the membrane was assessed by performing FRAP analysis. A circular area (radius, *w* = 1 μm) was bleached using high intensity illumination. The recovery kinetics of the lipophilic tracer molecule in this area was monitored. The recovery kinetics was fitted to an exponential of the form:(1)$$f\left(t\right)=A\left(1-e^{-t\tau }\right)$$
(2)$$\tau_{1/2}=\ln\left(\frac{0.5}{-\tau }\right)$$

Diffusion coefficient (*D*) was estimated by the relation below.
(3)$$D=0.88\frac{w^{2}}{4\tau_{1/2}}$$
where $\tau_{1/2}$ and *w* denote the half time of recovery and radius of uniform bleach laser respectively.

## 6. Measurement of Trans-Bilayer Movement

Trans-bilayer movement of symmetrically labeled NBD-PC in SULB was measured from the fluorescence quenching by using dithionite as reducing agent of NBD fluorophore. This fluorescence decline corresponds to the dithionite-mediated reduction of NBD analogs localized in membrane leaflets. To measure the flip-flop movement in our SULB system, we labeled the bilayer with NBD PC (0.25 mol%) during GUV preparation (POPC:CHOL (1:1)). Then, we ruptured NBD labeled GUVs (40 μL of 1 mg/mL GUV stock) on PCTE membrane by addition of 20 μL of 3M CaCl_2_. After removal of unfused GUVs, we added 20 μL of 1 M sodium dithionite (Na_2_S_2_O_4_). We measured the reduction of fluorescence intensity with time after addition of dithionite. To understand the effect of ClyA pore formation in our SULB flip-flop movement, before addition of dithionite, we treated our NBD labeled SULB with 100 nM ClyA.

## 7. Transfection and Generation of Virus Stock

Both BHK and Vero E6 cells were maintained in Dulbecco’s Modified Eagle Medium (DMEM, Gibco Thermo Fisher Scientific, Waltham, MA, USA), supplemented with 10% fetal bovine serum, 15 mM HEPES, 200 mM glutamine, 100 mM non-essential amino acids penicillin-streptomycin 100 U/mL at 37 °C, 5% CO_2_. The plasmid pFK-DV encoding a synthetic copy of full-length genomic sequence DENV2 strain 16681 was a gift from Dr. Ralf Bartenschlager. RNA was synthesized from the linearized plasmid by in vitro transcription (HiScribe™ SP6 RNA Synthesis Kit) followed by capping. RNA was purified by Sodium-acetate precipitation. The purified RNA was transfected to the BHK cells by lipofectamine MessengerMax (Thermo Fisher Scientific) and the supernatant was harvested and cleared from the cell debris by low-speed centrifugation. The virus was produced by infecting Vero E6 cells and the supernatant was collected and purified from cell debris by low-speed centrifugation. The presence of the virus was confirmed by immunocytochemistry using the pan flavivirus E specific monoclonal antibody 4G2 with 1:500 dilution (Catalog No. GTX57154) and secondary antibody of Goat anti-Mouse IgG (H + L) Cross-Adsorbed Secondary Antibody, Alexa Fluor™ 488 (Catalog No. A-11001) with the concentration of 1 µg/mL.

## 8. Virus Purification and Generation of DiD-Labeled Virus

The virus supernatant was collected 72 h post the infection and the virions were pelleted by ultracentrifugation at 4 °C in Beckman type 70 Ti rotor for 15 h for 30,000× *g*. The virus pellet was resuspended in a buffer containing 20 mM Tricine (N-(2-Hydroxy-1,1 bis (hydroxymethyl-ethyl-glycine) pH 7.8, 140 mM NaCl and 0.005% Pluronic F-127. The virus was further purified over an opti-ep density gradient (SW 32 Ti rotor, 32,000 RPM at 4 °C for 3 h) with 15%, 25%, 40%, and 60%. The band between 25% and 40% was harvested, aliquoted and stored at −80 °C. Prior to the experiment, the virus was collected and cleared from opti-prep using 100 kDa filter (Amicon ultra 100 kDa centrifugal filter unit). Approximately 3 × 10^6^ genome equivalent DENV Virus were mixed with DiD to a final concentration of 20 µM (of DiD). After an incubation of 10 min, the unincorporated dye was removed by gel filtration on a Sephadex G-25 column in Tricine buffer. DiD-labeled virus was stored at 4 °C and used within 2 days.

## Figures and Tables

**Figure 1 membranes-12-01190-f001:**
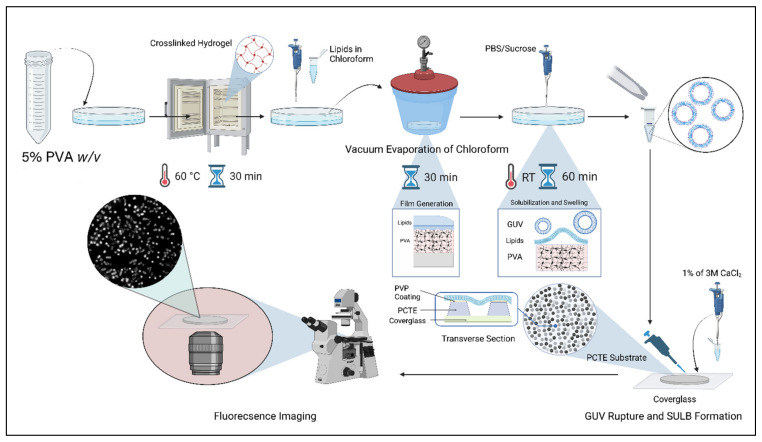
Preparation of suspended lipid bilayer on PCTE substrate. The schematic shows the procedure for SULB platform preparation. GUVs prepared with gel-assisted technique are ruptured and spread over PCTE polymeric substrate placed on a cover-glass.

**Figure 2 membranes-12-01190-f002:**
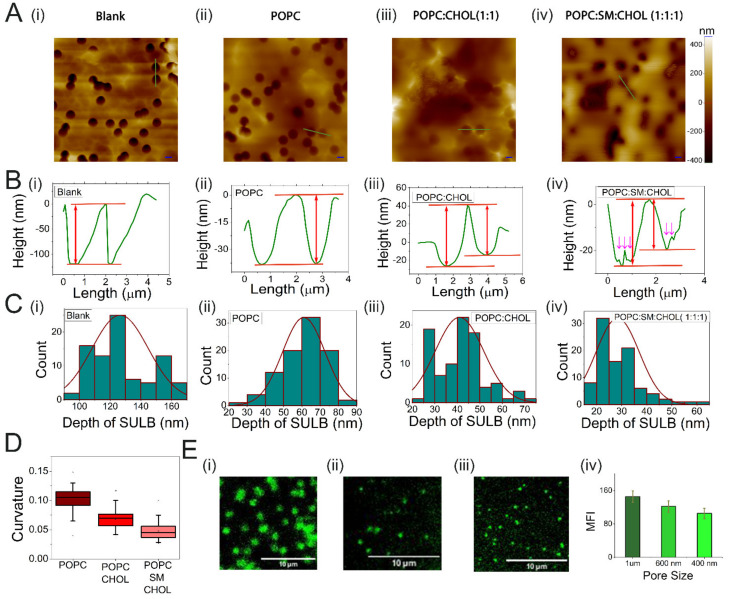
Characterization of SULB system. (**A**) AFM topographic images, (**B**) line profiles and (**C**) the distribution of the depth of the microwells for (**i**) Blank PCTE substrate, (**ii**) POPC SULB, (**iii**) POPC:CHOL (1:1) SULB, and (**iv**) POPC:SM:CHOL (1:1:1) SULB are shown. Scale bar = 2 μm in panel. The average size of the microwells was found to be 1.2 μm. (**A**) The arrows in (**B**(**iv**)) indicate the surface roughness in ternary Suspended bilayer which may appear due to the Lo/Ld phase separation. (**D**) Curvature distribution of SULB systems prepared with different lipid compositions. The error bar represents the standard deviation. The total number of cavities, *n* = 85 for blank, *n* = 91 for POPC, *n* = 91 for binary and *n* = 92 for ternary system are measured. (**E**) Fluorescence microscopy images of DiI labeled SULB system of microwell size (**i**) 1 μm, (**ii**) 600 nm, and (**iii**) 400 nm. (**iv**) Mean fluorescence intensity of DiI labeled suspended bilayer for different microwell size. Scale bar = 10 μm.

**Figure 3 membranes-12-01190-f003:**
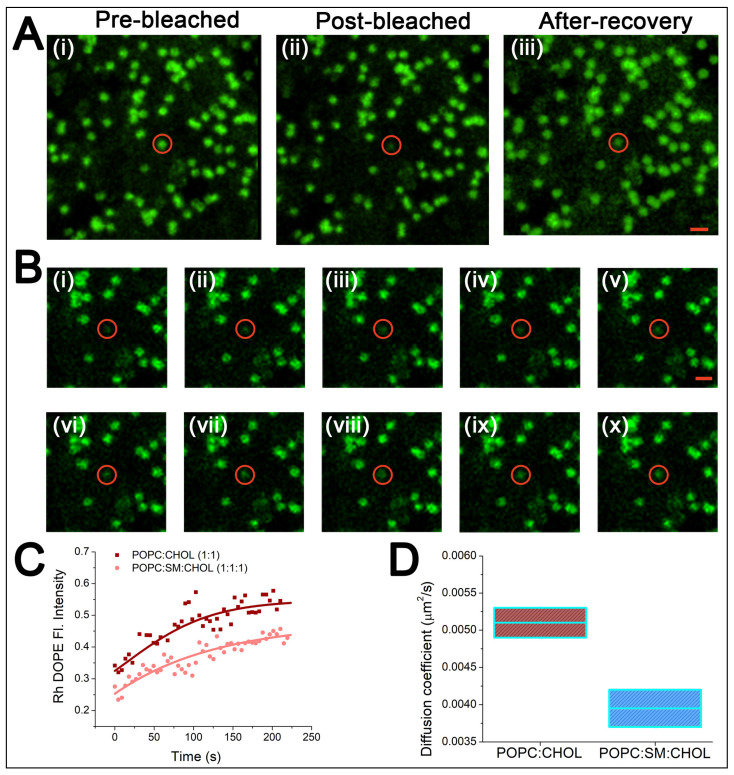
FRAP analysis of lipids in the SULB system. (**A**) Representative confocal microscopic images of RH-DOPE embedded in SULB systems composed of ternary lipid mixtures. (**i**) Prebleached, (**ii**) post-bleached and (**iii**) after recovery images are shown for the ternary SULB system. The red circle denotes the selected spot for bleaching. Scale bar = 2 μm. (**B**) Time lapse image sequences (**i**–**ix**) show the recovery of the selected bleached spot with an interval of 37 s. Here, we used ternary lipid mix to form the SULB system. (**C**) Fluorescence recovery curves for binary and ternary SULB systems show slow recovery rate for both but the recovery rate and fraction recovered for the binary membrane was marginally larger. This is expected because the abundance of Lo domain in ternary bilayer might restrict lipid mobility in the system. (**D**) Diffusion coefficient for RH DOPE in binary and ternary SULB systems. Each data represents the average of two individual experiments.

**Figure 4 membranes-12-01190-f004:**
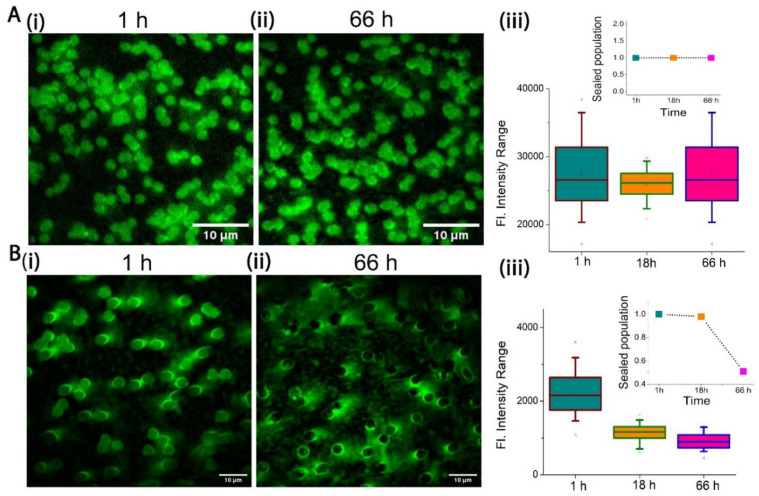
Stability of SULB system. (**A**) Fluorescent images of SRB (100 nM) entrapped binary SULB (POPC:CHOL, 1:1 on 1 μm microwells) after (**i**) 1 h and (**ii**) 66 h are shown (**iii**) SRB intensity distribution as measured from individual dye filled cavities as function of time is shown. The total number of cavities measured n = 163 for 1 h, n = 157 for 18 h and n = 167 for 66 h. The inset is showing the fractions of sealed cavities with time of incubation. (**B**) Fluorescent images and corresponding plots as panel A for 3 μm microwells is shown. The total number of cavities accounted is *n* = 70 for 1 h, *n* = 61 for 18 h and *n* = 56 for 66 h. The inset is showing the fractions of sealed cavities with time of incubation. (For this calculations, total number of pores accounted is *n* = 500 for 1 h, *n* = 479 for 18 h and *n* = 1588 for 66 h).

**Figure 5 membranes-12-01190-f005:**
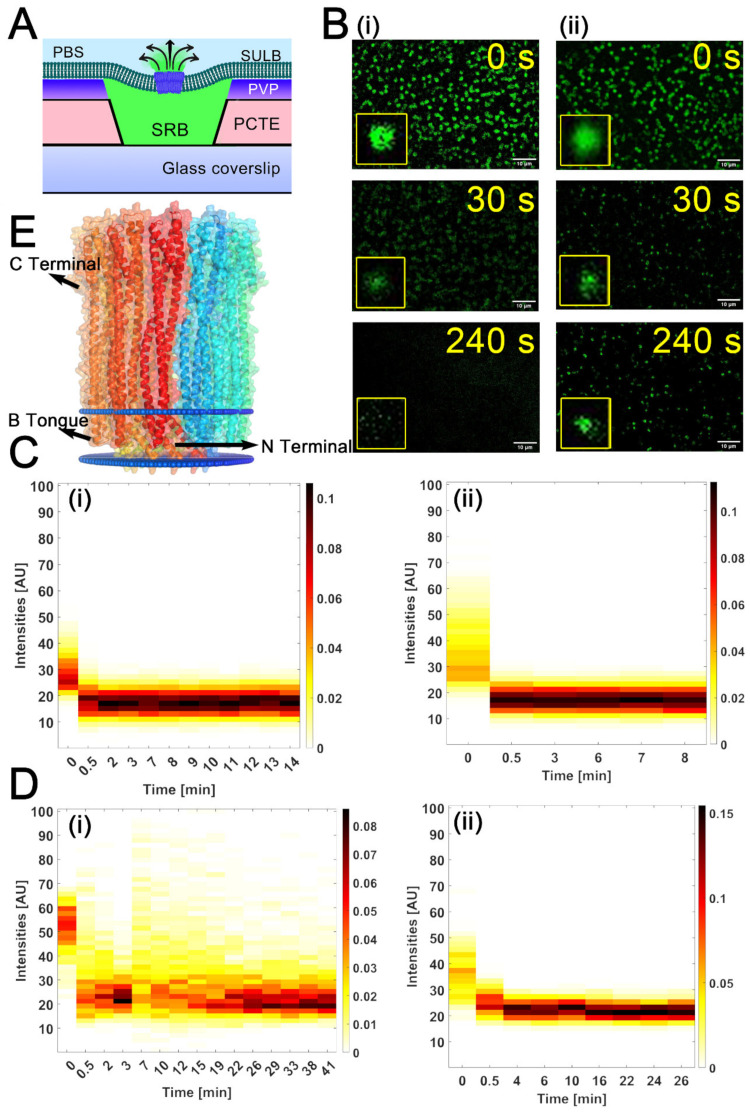
Pore formation in SULB by Cytolysin A. (**A**) Schematic shows the pore formation and associated SRB dye leakage from the SULB due to protein nanopore formation. (**B**) Representative fluorescence images of SRB entrapped SULB platform composed of (**i**) ternary and (**ii**) binary lipid compositions at different time points upon the addition of 100 nM ClyA are shown. (**C**) Heat maps of the temporal SRB intensity changes in 600 nm microwell for (**i**) binary and (**ii**) ternary membrane systems have been shown. (**D**) Heat maps of the respective SRB leakage from 1 μm micro-well size for (**i**) binary and (**ii**) ternary membrane systems. (**E**) Membrane embedded dodecameric pore structure of ClyA as obtained from OPM software.

**Figure 6 membranes-12-01190-f006:**
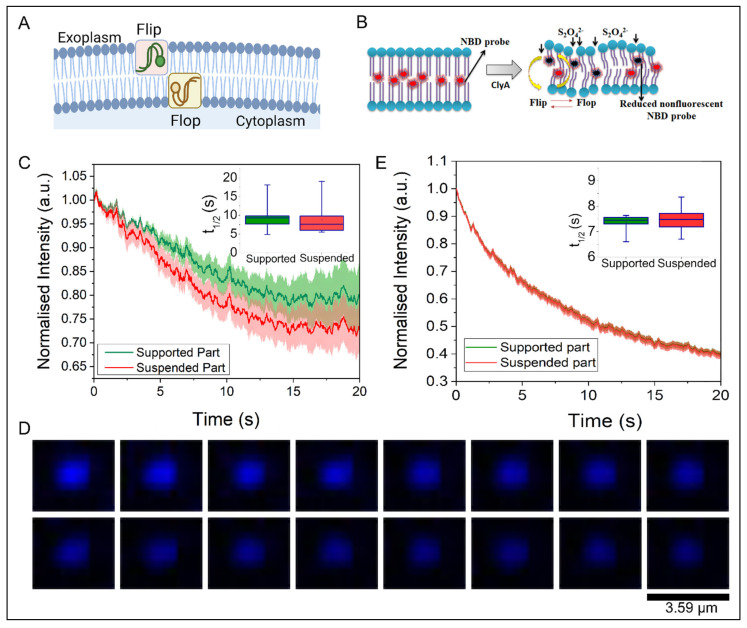
Lipid flip-flop on SULB. (**A**) Schematic depicts the trans-bilayer movement in terms of lipidic flip flop dynamics in bilayer. (**B**) Sodium dithionite reduction strategy for measurement of flip-flop motion in suspended bilayer system in absence and presence of ClyA. The quencher irreversibly reduces the NBD dye as it is exposed to the aqueous phase. (**C**) Normalized NBD fluorescence intensity in SULB was plotted against time to monitor the trans-bilayer movement. We measured the temporal NBD fluorescence intensity reduction due to addition of Na_2_S_2_O_4_ to the SULB systems in absence of ClyA. Inset is showing the half-lives of reduction of NBD signal for both suspended and supported parts. (**D**) Time lapse images of single suspended bilayer cavity, while being treated with ClyA and incubated for 30 min. Each image is separated by a time interval of 1.25 s. (**E**) The temporal NBD Fl. Intensity reduction due to addition of Na_2_S_2_O_4_ to the SULB systems in presence of ClyA. Inset is showing the half-lives of reduction of NBD signal for both suspended and supported segments.

**Figure 7 membranes-12-01190-f007:**
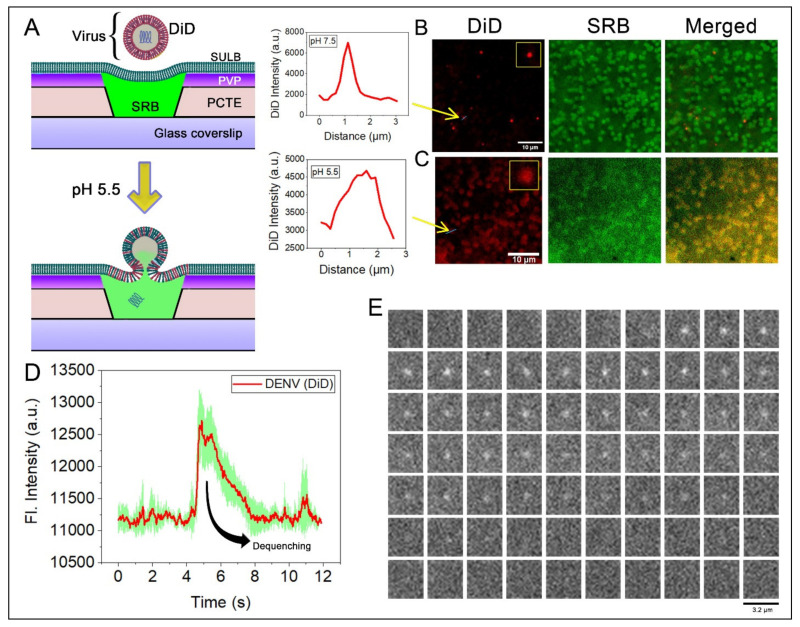
Dengue virus fusion on SULB platform. (**A**) Schematic of the experimental setup for the measurement of DENV fusion on the PCTE SULB system. DiD labeled DENV were added on SRB entrapped SULB platform made of POPC:SM:CHOL:POPG at equimolar concentration. After pH drop, fusion of the virus membrane ensues and the DiD diffuses into the target SULB membrane, leading to a dequenching of the signal along with the partial leakage of SRB from the microwell cavity. (**B**) Dual color imaging of our SULB platform-entrapped SRB dye and bound DENV (red lipid dye DiD) at pH 7.5. The arrow mark is showing the sharp Gaussian intensity distribution of DiD labeled DENV. (**C**) Dual color imaging of our SULB platform-entrapped SRB dye and bound DENV (red lipid dye DiD) at pH 5.5. The arrow mark is showing the broad intensity distribution of DiD labeled DENV. (**D**) Real-time single virus fusion event of DiD labeled DENV monitored on the SULB. The green shade depicts the standard deviation of 5 individual virus fusion events. (**E**) Movie panel showing DiD DENV particle binding and fusing on SULB. Each box represents the same region separated in time by 25 min.

## Data Availability

The data presented in this study are available on request from the corresponding authors.

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
