# Peer review of "A Versatile Suspended Lipid Membrane System for Probing Membrane Remodeling and Disruption"

_membranes, 2022, doi:10.3390/membranes12121190_

Round 1

Reviewer 1 Report

In this work, Rahul Roy and his co-authors developed a suspended lipid membrane platform for membrane remodeling applications. I am in support of publishing this work. However, the paper must address all the major issues before publication in Membranes.

1)     Suitably for Membranes, the paper is concentrated on lipid membranes and device development. However, it does not describe how the PCTE holey substrate was generated. This should be explained in some detail, possibly in supplementary materials.

2)     The paper is not sufficiently scholarly, mainly because prior work on the suspended lipid membrane from other groups (Claudia Steinem and Sathish Ramakrishnan) is not adequately acknowledged. There has been a lot of work on the suspended lipid membrane and its application to fusion (Biophysical Journal 119 (1), 151-161; Langmuir 34 (20), 5849-5859; Elife 2020, e54506, 2020; Biophysical journal 116 (2), 308-318; Biophysical journal 112 (11), 2348-2356; Sci Rep 5, 12006 (2015)) from suspended membranes that should be referenced.

3)     The authors present the GUV rupturing to form a suspended bilayer on the PCTE substrate. The microwell size of PCTE is 1 um, 600 nm, and 400 nm. If the GUV size is 10 mm, why is the bilayer not uniform and looks puncta? Fig 2E center image appears to have much noise. Why is that? Is this bilayer formed from the same GUV? The authors must discuss this in detail.

4)     Compared to suspended lipid membranes reported in the literature (Sci Rep 5, 12006 (2015);

Langmuir 2018, 34, 20, 5849–5859); the diffusion coefficient of lipids in this system is very low. Can you describe this behavior?

5)     The most significant advantage of suspended lipid membranes has excellent lateral diffusion of lipids and proteins. Did you check Protein Frap to validate this claim?

6)     Can you add the diffusion coefficient of POPC lipid? Cholesterol tends to form domain at higher concentrations, which slows down lipid diffusion.

7)     How the fluorescence intensity in Figure 4(B, III) is measured? It looks like there is no fluorescence at the center of the hole. Can you comment on this?

Author Response

Reviewer 1

General comment: In this work, Rahul Roy and his co-authors developed a suspended lipid membrane platform for membrane remodeling applications. I am in support of publishing this work. However, the paper must address all the major issues before publication in Membranes.

Comment 1:Suitably for Membranes, the paper is concentrated on lipid membranes and device development. However, it does not describe how the PCTE holey substrate was generated. This should be explained in some detail, possibly in supplementary materials.

Author's Response:Thank you for pointing this out and we regret the lack of details. We have used off-the-shelf Track Etched Polycarbonate (PCTE) substrate from STERLITECH as the base for our SULB platform. We have now incorporated a brief description of their production methodology into the method section of the paper.

Please refer to the text on page 21 in the revised version.

"The PCTE substrates were purchased from STERLITECH. For the generation of the PCTE substrates, a nonporous polycarbonate film is exposed to heavy ions in a particle accelerator, which leaves "tracks"as the polymer chains are ruptured and ionized while the ions traverse across the depth of the susbtrate. The polycarbonate film is then "etched" in a bath containing warm NaOH to form the pores. Control of temperature, pH and etching time determines the pore dimensions in the PCTE. "

Comment 2:The paper is not sufficiently scholarly, mainly because prior work on the suspended lipid membrane from other groups (Claudia Steinem and Sathish Ramakrishnan) is not adequately acknowledged. There has been a lot of work on the suspended lipid membrane and its application to fusion (Biophysical Journal 119 (1), 151-161; Langmuir 34 (20), 5849-5859; Elife 2020, e54506, 2020; Biophysical journal 116 (2), 308-318; Biophysical journal 112 (11), 2348-2356; Sci Rep 5, 12006 (2015)) from suspended membranes that should be referenced.

Author's Response:We thank the reviewer for the suggestions. Indeed, suspended membranes have been employed before. Here, we demonstrate a facile way of developing free-standing lipid bilayers that does not require nanofabrication.

Based on the reviewer comments, we have modified our references and included the references in page number 27 (references 26 through 31) in the revised manuscript.

Comment 3:The authors present the GUV rupturing to form a suspended bilayer on the PCTE substrate. The microwell size of PCTE is 1 um, 600 nm, and 400 nm. If the GUV size is 10 mm, why is the bilayer not uniform and looks puncta? Fig 2E center image appears to have much noise. Why is that? Is this bilayer formed from the same GUV? The authors must discuss this in detail.

Author's Response:Thank you for pointing this source of confusion. In figure 2E, we have shown the fluorescence images of DiI labelled suspended bilayer systems. We illuminate the sample from underneath the polymeric substrate and hence excitation of the bilayer not directly over the microwells is blocked. Additionally, the membranes are composed of POPC and Cholesterol in 1:1 ratio. Due to the presence of high cholesterol, the lipid molecules can form domains, and this heterogeneity might be responsible for this punctated conformation. We replaced the middle image of Fig 2E panel with a higher-quality image. Kindly refer to figure 2E and the text in the revised version.

“Due to the illumination of the bilayer from the bottom of the substrate, the DiI in the lipid bilayer not on the microwells is not completely excited providing enhanced contrast.”

Comment 4:Compared to suspended lipid membranes reported in the literature (Sci Rep 5, 12006 (2015);Langmuir 2018, 34, 20, 5849–5859); the diffusion coefficient of lipids in this system is very low. Can you describe this behavior?

Author's Response:For the FRAP experiment, we have selectively bleached only the suspended part over a microwell. Since, trapping of the water soluble dye indicates pinning of the bottom membrane, we suspect poor lipid diffusion is manifestion of poor exchange of lipids between the suspended section and the supported section of the membrane.

Previous reports indicate a similar restricted diffusion behaviour of Rhodamine DOPE in the suspended bilayer segments (Langmuir 2017, 33, 46, 13277–13283. https://doi.org/10.1021/acs.langmuir.7b02156). It has been shown that Lo domain preferably stays in the supported part and the pore rims ( https://doi.org/10.1073/pnas.1704199114). In contrast, the Ld phase is present in the suspended part. We suspect that the high concentration of cholesterol can lead to local domain formation in our SULB and these domains give rise to local confinement which could further lead to lower diffusivity of Rh DOPE in the suspended section.

Please refer to the updated text in page number 9.

Comment 5:The most significant advantage of suspended lipid membranes has excellent lateral diffusion of lipids and proteins. Did you check Protein Frap to validate this claim?

Author's Response:Thank you for your valuable comment. We understand that the excellent lateral diffusion of lipids and proteins is the advantage of a suspended lipid system but we haven't done Protein Frap since we have validated our system with various functional assays like cytolysin A pore formation and virus membrane fusion and the characteristic changes we observed in these assays are in good agreement with previously reported results.

Comment 6:Can you add the diffusion coefficient of POPC lipid? Cholesterol tends to form domain at higher concentrations, which slows down lipid diffusion.

Author's Response:We have measured the diffusion coefficient of Rh-DOPE lipid in our SULB system composed solely of POPC lipid. For this measurement, we have selectively chosen the whole suspended bilayer region over single microwell and we found very low diffusion coefficient (D= 0.0093 μm2/s) for POPC SULB. The pinning of the lipid bilayer on the supported segments and rim of the microwell might be the reason behind the restricted mobility of the lipid molecules from the supported to suspended region resulting in the low and slow recovery of the lipid molecules in the suspended region when we completely bleach all molecules in the free-standing segment.

Comment 7:How the fluorescence intensity in Figure 4(B, III) is measured? It looks like there is nofluorescence at the center of the hole. Can you comment on this?

Author's Response:Figure 4 reports on the SULB system's stability. We calculated the fluorescence intensity of each microwell and the change in fluorescence with time. The microwells were filled with SRB dye, and the bilayer is formed over it. Upon membrane disruption, the dye leaks out of the microwell and into the buffer solution that permeates the top part of the system. This leakage causes the intensity to drop in the center region of the microwell, yet there would be an intensity at the walls emanating due to non-specific binding of the dye to the microwell walls. This signature at the walls confirms that a dye filled microwell existed while a dark interior tells us that the pore is ruptured. This allows us to count the ratio of intact pores to total pores.

Reviewer 2 Report

Sannigrah et al, developed a new lipid membrane system that is suitable to test membrane properties such as clustering, curvature, stiffness, and remodeling and disruption processes. This system can be used for sensitive fluorescence microscopy studies which they show by measuring lipid diffusion (FRAP), membrane curvature and heterogeneity (AFM), membrane rupture (dye leakage), lipid flip-flop (NBD-lipid quenching), and membrane fusion (DiI dequenching). They were used to characterize pore formation by ClyA and dengue virus fusion. The paper is well written and the methodology is well explained. I just have a minor comment. Although the authors claim that the system can be used in a high-throughput fashion, they don’t actually prove it. I suggest either removing this assumption or demonstrating it experimentally. 

Author Response

Reviewer 2

General comment:Sannigrah et al, developed a new lipid membrane system that is suitable to test membrane properties such as clustering, curvature, stiffness, and remodeling and disruption processes. This system can be used for sensitive fluorescence microscopy studies which they show by measuring lipid diffusion (FRAP), membrane curvature and heterogeneity (AFM), membrane rupture (dye leakage), lipid flip-flop (NBD-lipid quenching), and membrane fusion (DiI dequenching). They were used to characterize pore formation by ClyA and dengue virus fusion. The paper is well written and the methodology is well explained.

Comment 1:I just have a minor comment. Although the authors claim that the system can be used in a high-throughput fashion, they don't actually prove it. I suggest either removing this assumption or demonstrating it experimentally.

Author’s response: We thank the reviewer for this suggestion. Our platform allows measurement of hundreds of micro-wells that act as individual free standing lipid membrane platforms, we agree similar other platforms do exist. Therefore, we have removed the term ’high-throughput’ in the revised version.

Round 2

Reviewer 1 Report

Publish as it is